# Effects of Pectinase Pre-Treatment on the Physicochemical Properties, Bioactive Compounds, and Volatile Components of Juices from Different Cultivars of Guava

**DOI:** 10.3390/foods12020330

**Published:** 2023-01-10

**Authors:** Xiaowei Chen, Yujuan Xu, Jijun Wu, Yuanshan Yu, Bo Zou, Lu Li

**Affiliations:** Sericultural & Argi-Food Research Institute, Guangdong Academy of Agricultural Sciences/Key Laboratory of Functional Foods, Ministry of Agriculture and Rural Affairs/Guangdong Key Laboratory of Agricultural Products Processing, No.133 Yiheng Street, Dongguanzhuang Road, Tianhe District, Guangzhou 510610, China

**Keywords:** guava juice, cultivar, pectinase treatment, bioactive compounds, flavor

## Abstract

In this study, the physicochemical properties, antioxidant capacity, and volatile compounds of the juices of different guava cultivars before and after pectinase treatment were evaluated. The results showed that the guava juice of the small fragrant (SF) cultivar exhibited the highest ascorbic acid concentration (1761.09 mg/L), and the highest contents of total phenolics (329.52 mg GAE/L) and total flavonoids (411.13 mg RE/L) were both found in the juice of the watermelon red (WR) cultivar. After pectinase treatment, the juice yield and the titratable acid, sugar components, total phenolics and total flavonoids, and antioxidant capacity levels of the guava juices were all higher than those of the non-pectinase group. However, lower sensory evaluation scores were obtained in the pectinase-treated guava juices. Aldehydes and terpenoids were the main flavor components in the guava juices, which were responsible for the aroma of the juice, while their relative contents were different in the four cultivar guava juices. Furthermore, pectinase treatment could change the amounts and relative contents of volatile compounds in the guava juice. During the pectinase treatment process, the relative contents of the main aroma constituents in the guava juices were significantly decreased. The findings of this research provide valuable information for the processing of guava juice.

## 1. Introduction

Guava (*Psidium guajava* L.) belongs to the Myrtaceae family, native to the American tropics [1]. It is a tropical fruit tree with strong adaptability, which is grown commercially all over the world, such as in China, India, Thailand, Malaysia, and Vietnam [2,3]. Guava enjoys a good reputation as a superfruit due to it is abundant nutritional and functional ingredients, such as various vitamins, minerals, phenolics, and dietary fibers [4,5]. As is well known, orange is a good dietary source of ascorbic acid, while guava has a threefold higher ascorbic acid content than oranges [6,7]. Meanwhile, guava has the antioxidant, antibacterial, anti-inflammatory, antihyperglycemic, immune regulation, and hepatoprotective abilities, and it has been used as a medicine to treat multiple diseases [4,8]. In addition to fresh consumption, guava is also processed into a variety of products, such as juice, jam, jelly, wine, and dried fruit [9]. Among them, guava juice has become an important economic alternative to carbonated drinks, caffeinated drinks, and other beverages because of its high nutritional value and rich aroma [10].

Nowadays, consumer interest in nutrient-rich fruits and fruit products is on the rise as the public becomes more aware of the impact of their diet on their health. However, the chemical composition content of the same fruit varied between the cultivars. According to a previous study, the amounts of nutrients in dates depend on the cultivar, with ‘Sokkery’ having the highest dietary fiber content [11]. The ascorbic acid and polyphenols content in amla fruit vary among cultivars and the highest levels were found in the ‘Chakaiya’ cultivar [12]. Meanwhile, the cultivar also has great influence on the flavor of the fruits and fruit products. The wine fermented by different kiwifruit cultivars exhibited great differences in flavor, with the highest aromatic components content being found in ‘Guichang’ wine [13]. Furthermore, some studies have also reported that the chemical composition contents of different guava cultivars vary [3,14], while detailed information about the processing properties and aromatic components of different guava cultivars is still scarce. Hence, investigating the processing characteristics and aromatic composition variation between guava cultivars could provide valuable information for the production of high-quality guava juice.

Pectinases are a class of enzymes that catalyze the decomposition of the substances contained in pectin, mainly including pectin esterase and depolymerases. The former can cleave the ester bonds in pectin to produce pectic acid, and the latter can cleave the glycosidic bonds in pectic acid [15]. Pectinase has been widely used in fruit and vegetable juice processing because of it has the advantages of degrading and removing pectin substances, increasing juice yield, protecting juice nutrients and improving production efficiency [16,17]. Pectinase has also been used for many years in guava juice processing. However, the current study mainly concentrates on optimizing enzymolysis conditions, the kinetics of enzymatic hydrolysis [18,19,20,21,22]. Few studies have compared the effect of pectinase treatment on the quality of guava juice. The influences of pectinase treatment and the cultivar on the physicochemical properties, bioactive compounds, and volatile aroma compounds of guava juice is not clear. Therefore, in this study, the effects of pectinase treatment on the physicochemical properties, ascorbic acid, individual sugar, sensory evaluation, antioxidant capacity, and volatile components of guava juice of different cultivars were investigated.

## 2. Materials and Methods

### 2.1. Enzyme and Reagents

Pectinase was obtained from Yantai Dibaos Brewing Machinery Co., Ltd. (Shandong, China). Ascorbic acid, fructose, glucose, sucrose, 1,1-diphenyl-2-picrylhydrazyl (DPPH), and 2,2’-Azinobis-(3-ethylbenzthiazoline-6-sulphonate) (ABTS^+^) were purchased from Shanghai Yuanye Biotechnology Co., Ltd. (Shanghai, China).

### 2.2. Plant Materials

Four common guava fruit cultivars—white pulp (cultivar 1: small fragrant fruit (SF); cultivar 2: white pearl (WP)), pink pulp (cultivar 3: four-seasons red (FR)) and red pulp (cultivar 4: watermelon red (WR))—were used in this study. These four cultivars were collected from Guava planting base of Guangzhou Chuangxian Agricultural Development Co., Ltd., Guangzhou, China. All of the guava fruits were harvested at the optimal harvest time (edible stage), and the judgment of guava maturity was according to that described by DINIZ et al. [23]. After harvesting, the guava fruits were stored at a low temperature (4 °C) and transported to our laboratory within 12 h.

### 2.3. Preparation of Guava Juices

The guava fruits were washed with distilled water, then the guava fruits were cut into small pieces and the pulp was extracted using a household juicer (LLJ-206J; Bestday Co., Ltd., Jiangmen, Guangdong, China). Subsequently, the guava pulp was divided into two groups. To one group 0.05% (*m*/*m*) pectinase was added to the guava pulp and it was soaked at 50 °C for 1 h (pectinase treatment, PT; the four cultivars were expressed as P-FR, P-SF, P-WR, and P-WP, respectively). The other group was the group without pectinase treatment (non-pectinase treatment, NT; the four cultivars were expressed as N-FR, N-SF, N-WR, and N-WP, respectively). The flow chart of the preparation of P-FR, P-SF, P-WR, and P-WP and N-FR, N-SF, N-WR, and N-WP guava juice is presented in Figure 1. All of the filtered juice samples were transferred to clear color sample bottles and then stored at −80 °C before analysis.

### 2.4. Determination of Juice Yield

The juice yield was calculated using the following Formula (1):Juice yield (%) = Yc/Y_0_ ×100(1)
where Yc and Y_0_ are the weight of juice and pulp (g), respectively.

### 2.5. Determination of Relative Turbidity and Juice Sedimentation

The relative turbidity and sedimentation indexes were measured using the method reported by Wu et al. [24] with some modifications. The relative turbidity (T) was calculated by the following Formula (2):T (%) = T_c_/T_0_ × 100(2)
where T_0_ and T_c_ are the turbidity values of the guava juice before and after centrifugation, respectively.

The sedimentation index (%, *w*/*w*) was expressed as the weight ratio of the centrifugal sediment to the juice sample.

### 2.6. Total Soluble Solids (TSS) and Titratable Acidity (TA)

TSS was measured using a digital refractometer (RP-101, Atago, Tokyo, Japan). The TA content of the guava juice was determined by the acid–base titration method. The volume of consumed NaOH was converted to gram of citric acid per liter.

### 2.7. Measurement of Color Parameters

The color analysis was performed at room temperature (25 °C) using a color difference meter (Hunter Lab Co., Reston, VA, USA) in the reflectance mode. Color was expressed in *L* for the brightness value, *a* for red–green value, and *b* for yellow–blue value.

### 2.8. Determination of Total Phenolics

The total phenolic content (TPC) was measured using the Folin–Ciocalteu method [25]. Five g of guava juice was mixed with 10 mL (MeOH:H_2_O:HCl = 80:19:1), and the mixture was treated with ultrasound for 20 min. Moreover, the supernatant was also used to determine the DPPH and ABTS capacity. The results were expressed as the milligram gallic acid equivalent (GAE) per liter of juice (mg GAE/L).

### 2.9. Determination of Total Flavonoids

The total flavonoids content was determined according to the colorimetric method from the previous report [24]. The total flavonoids content of guava juices was expressed as milligram rutin equivalent (RE) per liter of juice (mg RE/L).

### 2.10. HPLC Analysis of Ascorbic Acid and Sugar Component

The ascorbic acid of guava juice was extracted with 0.3% metaphosphoric acid, then determined by HPLC-DAD. The detection wavelength of ascorbic acid was 245 nm, and the results were expressed as mg/L. The analysis of sugars (fructose, glucose, and sucrose) by HPLC–ELSD was performed according to a previous study [26]. The results were expressed as the g sugar standard substance equivalent per liter of juice.

### 2.11. Analysis of Antioxidant Capacity

#### 2.11.1. DPPH Radical Scavenging Capacity Assay

The DPPH radical scavenging capacity of guava juice was determined by spectrophotometry according to the method reported by Zhang et al. [27] with minor modifications. A Trolox solution was used for calibration and the results were expressed as mg of Trolox equivalents per mL of juice (mg TE/mL).

#### 2.11.2. ABTS Radical Scavenging Capacity Assay

ABTS radical scavenging activities were determined by the method of Zou et al. [28] with some modifications. The results were expressed in mmol Trolox equivalents (TE)/L of juice.

### 2.12. Sensory Evaluation

The sensory evaluation was based on the methods of Sun et al., with slight modification [29]. The guava juice samples were placed randomly into clear glass bottles and then provided to each of the 20 trained panelists for sensory analysis. They were asked to drink water to clean their palate between the evaluation of each attribute. Five different attributes (taste, color, sweet and sour suitability, smell, and overall acceptability) of the guava juice samples were evaluated. Each attribute has a maximum of 20 points, and the total score of the juice is 100 points. The higher the score, the greater the intensity of the taste or the attribute in the sample.

### 2.13. Analysis of Volatile Compounds

The volatile compounds of the guava juices were extracted and measured using a headspace SPME GC-MS method according to Liu et al. [30]. The retention index (RIS) of unknown compounds was determined by the n-alkane (C5-C30) injection method. The peak area normalization method was used to calculate the relative contents of various volatile compounds.

### 2.14. Statistical Analysis

All of the experiments were repeated at three times. The results were expressed as the mean ± standard deviation. SPSS 23.0 software (Chicago, IL, USA) was used for the one-way analysis of variance (ANOVA) and Duncan’s multivariate test (*p* < 0.05). Pearson correlation was used to express correlations among bioactive compounds (total phenolics, total flavonoids, and ascorbic acid) and antioxidant activity.

## 3. Results and Discussion

### 3.1. Physicochemical Properties

Juice yield is one of the important indexes to determine whether the fruit is suitable for processing into juice [31]. From an economic perspective, low juice yield increases production costs, hence manufacturers commonly need to study fruit juice yield before fruit processing [32]. As shown in Table 1, the juice yield of guava varied significantly from 60.92% to 76.13% for the different varieties. Among these four guava varieties, the highest juice yield was obtained in ‘watermelon red’ (WR) (67.53%), followed by ‘four-seasons red’ (FR) (65.61%), ‘white pearl’ (WP) (64.99%), and ‘small fragrant fruit’ (SF) (60.92%). Moreover, the juice yield of the guava samples treated with pectinase was higher than that of the group without pectinase treatment. After pectinase treatment, the FR cultivar showed the highest juice yield, which was 15.17% higher than its corresponding non-pectinase-treated fruit juice. This result was consistent with previous research on the increase in juice yield after the pectinase treatment of banana, jujube, raspberry, soursop, and apricot [32,33,34,35,36]. This increase may be attributable to the breakdown of pectin and other cell wall components by pectinase, thereby promoting release of juice [31].

Turbidity is one of the significant parameters to evaluate the stability and sensory scores of fruit juice. Pectin, as a binder, is able to increase viscosity and thus increase the cloudiness, which is the principal reason for the haziness in fruit juice [37]. The influences of pectinase treatment on the relative turbidity of different cultivar guava juices are presented in Table 1. Compared to non-pectinase-treated guava juice, pectinase treatment significantly decreased the relative turbidity of guava juice. During the enzymatic treatment, pectinase hydrolyzes pectin molecules and promotes the formation of protein–pectin complexes, eliminating these colloidal particles in the juice and helping to reduce the turbidity of the juice [36]. Analogous phenomena also occur in other fruit juice, such as mango juice, watermelon juice, and apricot juice [32]. In addition to pectinase, the relative turbidity of guava juice was also affected by the cultivar. Among the four non-pectinase-treated guava juices, N-FR juice had the highest relative turbidity (31.96%), while the lowest relative turbidity was found in N-WP juice (12.92%). The increase in turbidity might owe to the presence of a high concentration of pectin, which made the juice cloudy [38].

The sedimentation index is another vital parameter to evaluate the stability and sensory scores of fruit juice, affecting the acceptance of consumers. Among these samples without pectinase treatment (Table 1), different varieties of guava juices had different sedimentation indexes, in which the sedimentation index of FR was the highest (53.54%) coupled with the sedimentation index of RP being the lowest (5.13%). Compared to non-pectinase-treated guava juices, the sedimentation indexes of the pectinase-treated guava juice all decreased. The sedimentation indexes of guava juices treated with pectinase decreased by 90.72% (N-FR), 82.08% (N-SF), 26.23% (N-WR), and 50.15% (N-WP), respectively. This result might be because of the reduction in cluster formation during the enzymatic treatment, thus enhancing the separation during filtration [32].

The total soluble solids (TSS) of fruit juice are commonly used as a quality control indicator in industry [39]. As described in Table 1, the significant differences were verified in the TSS of guava juices from different varieties. Among the non-pectinase treatment group, the highest TSS value was observed in N-FR juice (9.02°Brix), while the lowest value was measured in N-RP juice (7.71°Brix). Meanwhile, the TSS values of these four varieties of guava juices treated with pectinase were all increased in comparison with those of the non-pectinase treatment group. This increase could be explained by the fact that the action of pectinase on pectin substances in juices, resulting in the hydrolysis of these substances and the release of soluble components [40]. These results were consistent with those of other fruit juices [37,41,42].

Titratable acid (TA) is a meaningful determinant of juice taste and consumer acceptability. In this study, the TA content of different guava juices was also analyzed, and the results are illustrated in Table 1. In a comparison of the TA content of these four guava juices without pectinase treatment, the highest TA content was detected in N-WP juice (3.54 g/L), whereas the lowest TA concentration was found in N-SF juice (2.53 g/L). This difference was mainly due to the difference in the cultivars. Moreover, the TA content of guava juice was significantly increased by the addition of pectinase. In general, there is a certain correlation between pH and TA; the lower the pH value, the higher the TA content. A previous study reported that pectinase hydrolysis reduced the pH value of guava juice, which is in agreement with our findings [18]. The increase in TA content after pectinase treatment may be caused by the release of carboxyl (acid group) and ascorbic acid and the increase in galacturonic acid production [20].

Sweetness is one of the attributes that influences consumer satisfaction with fruit juices, and the intensity of the sweetness depends on their individual sugar content. Therefore, the sugar component (fructose, glucose, and sucrose) concentrations of different guava juices was analyzed. As can be seen in Table 2, the reducing sugar (fructose and glucose) and non-reducing sugar (sucrose) contents of guava juice from the four cultivars were significantly different. Among the non-pectinase-treated guava juices, N-WR juice obtained the highest reducing sugar content, while N-FR juice had the highest sucrose content. These results suggested that the varietal differences led to differences in the sugar component concentrations of guava juices. In addition, pectinase treatment also had a great effect on the individual sugar content of guava juice. Compared to pectinase-treated guava juice, non-pectinase treated guava juice had a higher non-reducing sugar concentration coupled with a lower reducing sugar content. After pectinase treatment, the content of reducing sugar in the guava juice increased while its non-reducing sugar content decreased, which was attributed to the action of pectinase on the polygalacturonse chain and the hydrolytic conversion of non-reducing sugars [32].

Color is an important sensory index of fruit juice, which has a great impact on acceptability by consumers. As shown in Table 2, the colors of the guava juices from different cultivars were markedly different. Among the non-pectinase-treated guava juice, the lightness (*L*) value showed an opposite trend to the turbidity, and the highest *L* value was found in N-WP juice. Compared to the guava juices without pectinase treatment, the *L* values of the pectinase-treated guava juices, except P-WP juice, were higher, indicating that pectinase treatment improved the lightness of the guava juices. Similar changes have been reported for apricot juice, jamun juice, and papaya juice [43]. The increased lightness of the juice could be due to pectinase decomposing the structural tissues of guava and releasing colored components [32]. Furthermore, the release of these colored ingredients also affects the *a* value of the juice. Among the non-pectinase-treated samples, the *a* values of N-FR and N-WP juice were positive, whereas the *a* values of N-SF and N-WP juice were negative. Compared to the N-FR juice, the *a* value of P-FR juice decreased, revealing that its pink color became lighter. Different from FR, the *a* values of the other guava juices treated with pectinase all increased, indicating that the greenness of the juice samples significantly decreased. This difference might be caused by the difference in the cultivars, while the specific mechanism needs to be further studied. Liberatore et al. also demonstrated that the color parameters of apple juice were significantly influenced by the cultivar and the processing method [44]. In contrast to the increased lightness of the juice, these colored components released by enzymatic hydrolysis resulted in a decrease in the *b* value. The *b* values of the guava juices without pectinase treatment were all higher than those of the pectinase-treated guava juices, and the highest *b* value was observed in the guava juices of the WR and WP cultivars. These results show that the yellowness of the guava juices treated with pectinase decreased evidently, while the blueness increased markedly. Hence, it could be concluded that the difference in the *b* values of the guava juices were mainly dependent on the variety and the pectinase treatment.

### 3.2. Bioactive Compounds

Ascorbic acid is the most abundant antioxidant in guava, which is essential for human health [45]. Hence, the ascorbic acid contents of different guava juices were measured (Figure 2A). Among the non-pectinase-treated samples, N-SF juice obtained the highest ascorbic acid content (1761.09 mg/L), followed by N-WR juice (1456.77 mg/L) and N-WP juice (1082.38 mg/L), while the lowest ascorbic acid concentration was observed in N-FR juice (913.08 mg/L). This result suggests that the ascorbic acid concentration of the guava fruit was greatly impacted by its cultivar, which is in line with previous research [3]. Furthermore, a significant decrease was found in the ascorbic acid content between the pectinase-treated guava juice and the non-pectinase-treated guava juice. This decrease is related to the low thermal stability of ascorbic acid and its susceptibility oxidative reactions during enzymatic hydrolysis [32].

Phenolics have been reported to have attracted interest owing to their therapeutic potential, especially in anti-cancer, anti-inflammatory, hypolipidemic, and hypoglycemic fields. Guava is rich in nutrients and has been confirmed to contain a variety of phenolics [46]. In the present research, the changes in the total phenolics of the guava juices of different cultivars were measured, and the results are presented in Figure 2B. Among the non-pectinase-treated guava juices, the highest total phenolics content (329.52 mg GAE/L) was found in N-WR juice, while the lowest total phenolics content was in N-WP juice. Compared to non-pectinase-treated guava juice, there was a substantial increase in the total phenolics content of guava juice after pectinase treatment. Notably, after being treated with pectinase, the total phenolics content in the guava juice of the SF cultivar increased the most, by 33.16%. This result was similar to previous studies on pectinase-treated papaya juice [43]. In another study, the use of pectinases increased the content of polyphenols in apricot juice [32]. The increase in total phenolics content might be due to the use of a 50 °C water bath during enzymatic hydrolysis, which facilitated the inactivation of polyphenol oxidase, peroxidase, and lipoxygenase and promoted the activity of pectinases, pectinesterases, and cellulases [15].

Guava fruit is reported to have a high content of total flavonoids, which is one of the antioxidant compounds [47]. Figure 2C showed the total flavonoids content of different guava juices. Among the non-pectinase-treated samples, N-WR juice possessed higher levels of total flavonoids than the other non-pectinase-treated guava juices. The total flavonoids contents of the pectinase-treated guava juices were all higher than those of the non-pectinase-treated guava juices. The total flavonoids contents of the pectinase-treated guava juices ranged from 541.59 mg RE/L to 649.92 mg RE/L, with the descending order of P-SF > P-WR > P-FR > P-WP. The significant increase in total flavonoids content after enzymatic treatment was associated with a higher degree of tissue rupture and the release of flavonoids from the peel cell wall. This finding is similar to the previous study on papaya juice treated with pectinase, where the addition of pectinase could effectively maintain the total flavonoids content in papaya juice [43].

### 3.3. Antioxidant Capacity

Guava is rich in many antioxidant substances, which are closely related to its antioxidant capacity [47]. Hence, the antioxidant capacities of different guava juices were measured by DPPH and ABTS assays. Among the guava juices treated without pectinase (Figure 3A), N-SF juice possessed the highest level of DPPH (452.91 mg TE/mL), followed by N-WR (418.96 mg TE/mL), N-FR (350.59 mg TE/mL), and N-WP juice (346.73 mg TE/mL). After pectinase treatment, the antioxidant capacity (DPPH) of the guava juices was enhanced, and there was a similar trend to that in the non-pectinase treatment group, namely, the order of the antioxidant activity (DPPH) of the guava juices was found to be P-SF > P-WR > P-FR > P-WP.

As depicted in Figure 3B, in the non-pectinase-treated group, N-WR juice showed the highest content of ABTS, followed by the N-FR, N-SF, and N-WP juices. The ABTS radical scavenging capacity of the guava fruit juices treated with pectinase were markedly higher than that of its corresponding non-pectinase-treated juices. A marked increase in the antioxidant activity of guava juice after pectinase treatment was demonstrated in the DPPH and ABTS assays. This increase might be attributed to an increase in the antioxidants (phenolics and flavonoids) decomposed from the cytoplasm [32]. Pectinase promotes the degradation of cell walls, thus releasing polyphenols and flavonoids which are localized in the cells [48]. Since these bioactive ingredients possess antioxidant activity, their increase may have improved the antioxidant potential of the guava juice. Analogous findings were also observed in raspberry and apricot juice [49]. In addition, Pearson correlation analysis was used to evaluate the correlation between the antioxidant components and antioxidant capacity of the guava juices, and the results are presented in Figure 4. The Pearson’s correlation coefficient measures a linear correlation. In general, the correlation coefficient is 0.8–1.0, it indicates an extremely strong correlation; 0.6–0.8, indicating a strong correlation; 0.4–0.6, indicating a moderate correlation; 0.2–0.4, indicating a weak correlation; 0.0–0.2, indicating a very weak correlation or no correlation. The total phenolics content was highly correlated with the antioxidant capacity determined by DPPH and ABTS assay, with the correlation coefficients being higher than 0.8. A medium correlation was observed between the total flavonoids content and the antioxidant activity, and the correlation coefficient was above 0.5. The antioxidant activity of the guava from the three cultivars from Thailand was also strongly correlated with the phenolics and flavonoids contents [3]. Similar to the total phenolics and total flavonoids, the ascorbic acid level was also correlated with the antioxidant activity. The antioxidant activity of papaya juice was also highly positively correlated with its ascorbic acid content [43]. From these results, it could be concluded that the antioxidant capacity of the guava juices was not only related to phenolics, but also to flavonoids and ascorbic acid, which might play a synergistic role in preventing damage to biological macromolecules by free radicals.

### 3.4. Volatile Compounds

Flavor is one of the sensory characteristics that affects the overall quality and consumer acceptance of juice [50]. The volatile compounds composition of different guava juices was analyzed by GC-MS. Principal component analysis (PCA) was used to evaluate the correlation of volatile components in different guava juices, and the results are shown in Figure 5. The sum of the cumulative variance contribution rates of the two principal components was 75.4%, with the PC1 at 59.9% and the PC2 at 15.5%. There was a clear separation among the different cultivar guava juices, showing that there were significant differences in the volatile components of guava juices from different cultivars. Meanwhile, compared with the non-pectinase treatment group, all of the pectinase-treated guava juices exhibited significant differences in volatile substances except for the WP cultivar. The above results indicate that both the cultivar and the pectinase treatment had great impacts on the volatile compounds of the guava juices.

To further analyze the effects of the different cultivars and pectinase on the volatile components of the guava juices, the relative contents of aroma compounds were measured (Figure 6 and Appendix A). A total of 83 flavor substances were identified, including 15 aldehydes, 15 alcohols, 11 esters, 32 terpenes, 5 ketones, and 5 other compounds. Among the non-pectinase-treated samples, N-FR juice had the largest number (40) of volatile compounds, followed by N-WP (38), N-WR (35), and N-SF (30) juice. Dursun et al. [50] stated that a greater number of volatile compounds was identified in the hawthorn fruit that they studied than those hawthorn fruit from cultivars ‘C. viridis’, ‘C. azarolus’, ‘C. pallasii’, ‘Waibahong’, ‘Damianqi’, and ‘Dajinxing’, which may be related to the cultivars. In addition to the number of volatile components, the relative contents of volatile components of guava juice were also different among the non-pectinase-treated groups. Aldehydes were the dominant volatile component in N-FR (52.84%) and N-SF (60.91%) juices, while terpenes were the main volatile component in the N-WR (83.09%) and N-WP (65.71%) juices. Aside from terpenes and aldehydes, the N-FR and N-SF juices also contained alcohols and esters. Notably, alcohols were also detected in the N-WR and N-WP juices, while no esters were detected. The most abundant aldehydes in the non-pectinase-treated guava juices were hexanal and €-2-hexenal, which provided fruit aromas such as apple and strawberry with green, herbal, and woody scents [51,52]. For the terpenes, 1-caryophyllene was considered as the dominant aroma compound of guava juice with a light clove-like fragrance [53]. Compared to non-pectinase treated guava juices, the number of volatile compounds increased in the pectinase-treated group and varied by the cultivar, indicating that pectinase treatment and the cultivar could affect the flavor composition of the guava juices. Zhu et al. [54] observed that the changes in the volatile components in the apple juice were closely associated with the varieties and the different processing processes, which was consistent with our findings. After treatment with pectinase, the relative contents of hexanal and (E)-2-hexenal in the P-FR and P-SF juices decreased, whereas they increased in the P-WR and P-WP juices. A similar phenomenon was found in apple juice, where the contents of hexanal and (E)-2-hexenal were reduced in the juice of ‘Golden Delicious’ and ‘Fuji’ varieties after enzymatic treatment, while the (E)-2-hexenal content increased in the ‘Ralls’ juice [54]. In contrast to the trend for aldehydes, there was an increase in the relative content of 1-caryophyllene in the P-FR and P-SF juices coupled with a decrease in the P-WR and P-WP juices. In this sense, this deviation in the relative contents might be due to the cultivar and pectinase hydrolysis [55]. Additionally, pectinase treatment increased the diversity and relative content of esters in the P-FR and P-SF juices, and the relatively high proportion of flavor components in the guava juices of the FR and SF varieties changed from aldehydes to aldehydes and esters. Meanwhile, after pectinase treatment, the relative content of esters in the guava juices of the WR and WP cultivars increased to 0.96% and 1.32%, respectively. Among these detected esters, leaf acetate and 3-phenylpropyl acetate accounted for a high proportion [56]. Although the diversity of the flavor components in the guava juices increased after pectinase treatment, their major aroma components content reduced, and their sensory evaluation scores were lower. The obvious aroma loss after pectinase treatment might be attributed to the volatile constituents released from ruptured guava pulp cells during enzymatic hydrolysis. The above results indicate that the relative contents of the primary aroma components in the guava juices were related to the cultivar and whether it was treated with pectinase. Pectinase treatment would lead to a marked loss of the major aromatic composition of the guava juices via degradation or volatilization.

### 3.5. Sensory Evaluation

In addition, a sensory evaluation on the color, flavor, sweet and sour suitability, taste, and overall acceptance of different guava juices was conducted (Appendix A). As shown in Figure 7, the sensory properties of different guava juices were significantly different. Among the non-pectinase-treated groups, the guava juice from the FR cultivar had the highest total score, which was mostly due to its optimal flavor attributes and sweet–sour suitability. In comparison, the guava juices after pectinase treatment showed lower scores in sensory attributes, especially in terms of flavor. Previous research has also reported that the flavor of the juices changes greatly after pectinase treatment [54,57]. This might be related to the conversion of volatiles into non-volatiles during pectinase treatment. However, the reaction mechanisms of these aroma changes during the enzymatic hydrolysis of the juice need to be further investigated. Furthermore, compared with the non-pectinase-treated group, the pectinase-treated guava juices had an obvious sour taste, which was consistent with its titratable acid and pH results (Table 1). Ninga et al. [18] also observed a significant reduction in the pH value of the pectinase-treated guava juices. In general, there was a remarkable difference in the overall acceptance of different guava juices, with the N-FR juice having the best acceptability, followed by the N-SF juice.

## 4. Conclusions

There is limited information on the effects of the cultivars and pectinase treatment on the overall quality of the guava juice. Hence, this study aimed to investigate the effects of the cultivars and pectinase treatment on the physicochemical parameters, bioactive compounds, and flavor components of the guava juice. Among the non-pectinase treatment groups, N-SF juice showed the highest content of ascorbic acid (1761.09 mg/L), while N-FR juice had the highest TSS, sugar composition content, and sensory evaluation score. Although a higher juice yield, content of titratable acid, sugar component, total phenolics and total flavonoids, and antioxidant capacity were found in guava juice treated with pectinase, its sensory evaluation scores were lower than the non-pectinase treated guava juice. Moreover, a close correlation between bioactive compounds (total phenolics, total flavonoids, and ascorbic acid) and antioxidant activity was found. Furthermore, a total of 83 volatile substances were detected from different guava juices. Aldehydes were the main contributors to the aroma of N-FR and N-SF juices, while the major flavor components of N-WR and N-WP juices were terpenes. After pectinase treatment, the amounts and relative contents of volatile components in guava juice were changed and their aromas were weakened. In combination, from the results of the physicochemical characteristics, sensory evaluation, and flavor compounds, it could be concluded that the WR and WP cultivars were suitable for processing into clear juice after pectinase treatment, while the FR and SF cultivars were suitable for directly processing into turbid juice.

## Figures and Tables

**Figure 1 foods-12-00330-f001:**
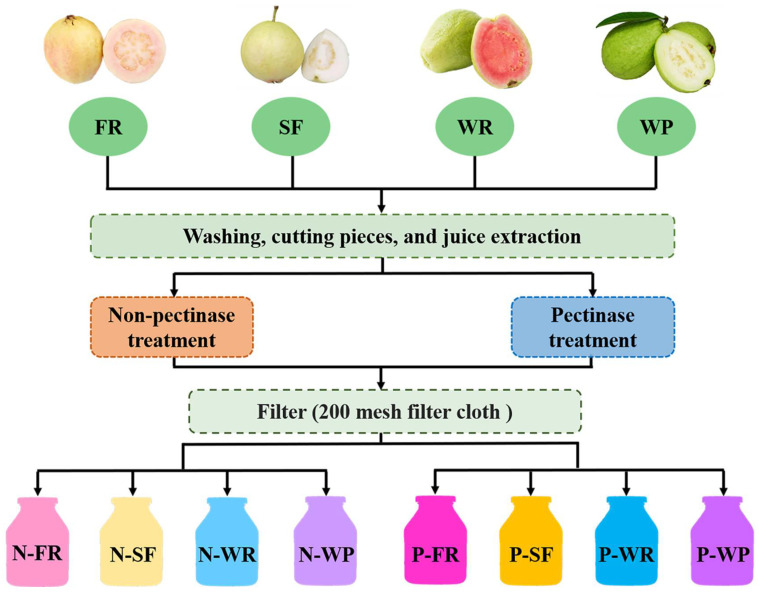
The flow chart of the preparation of P-FR, P-SF, P-WR, and P-WP and N-FR, N-SF, N-WR, and N-WP guava juice.

**Figure 2 foods-12-00330-f002:**
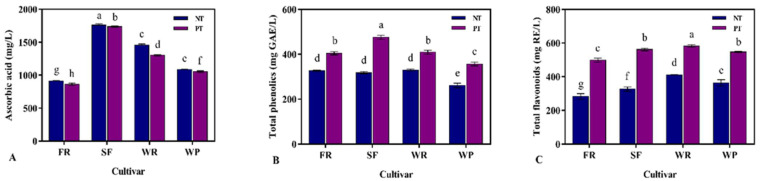
Effects of different cultivars and the effect of whether they were treated with pectinase or not on the ascorbic acid (**A**), total phenolics (**B**), and total flavonoids (**C**) of the guava juices. PT: pectinase treatment; NT: non-pectinase treatment. In each figure, different letters represent significant differences (*p* <0.05).

**Figure 3 foods-12-00330-f003:**
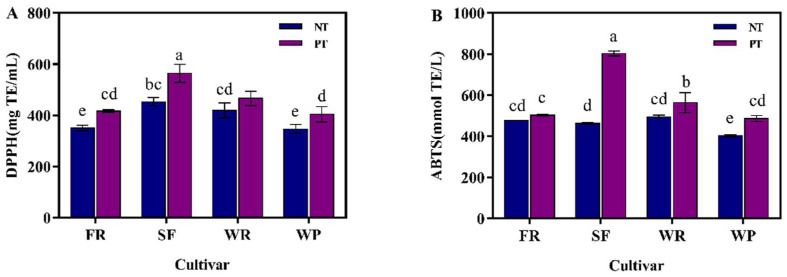
Effects of different cultivars and the effect of whether they were treated with pectinase or not on the DPPH (**A**) and ABTS (**B**) antioxidant capacity of the guava juices. PT: pectinase treatment; NT: non-pectinase treatment. In each figure, different letters represent significant differences (*p* <0.05).

**Figure 4 foods-12-00330-f004:**
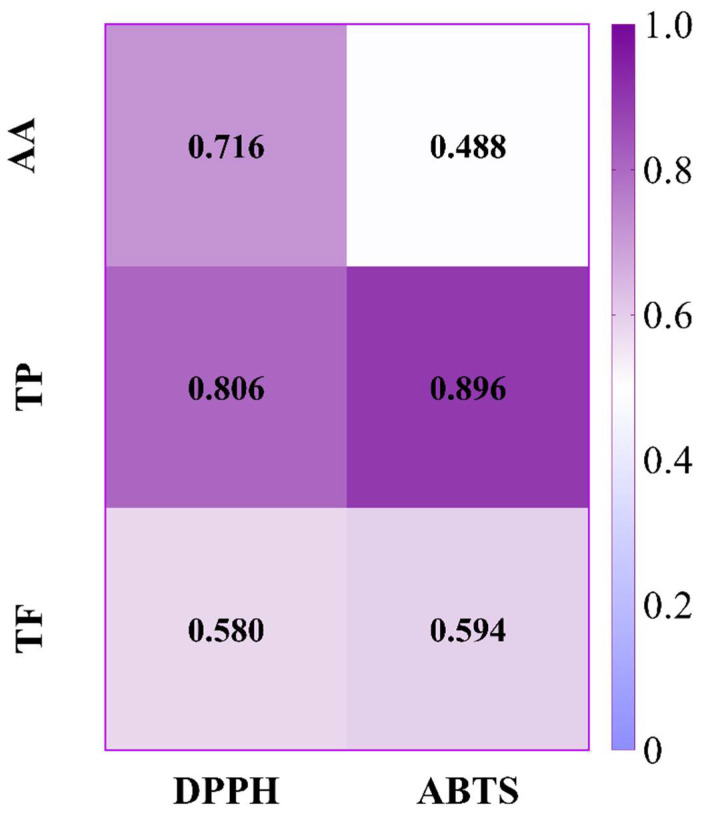
Pearson’s correlation coefficient between antioxidant capacity and the total phenolic, total flavonoid and ascorbic acid contents of guava juices. TP: Total phenolics; TF: Total flavonoids, AA: ascorbic acid.

**Figure 5 foods-12-00330-f005:**
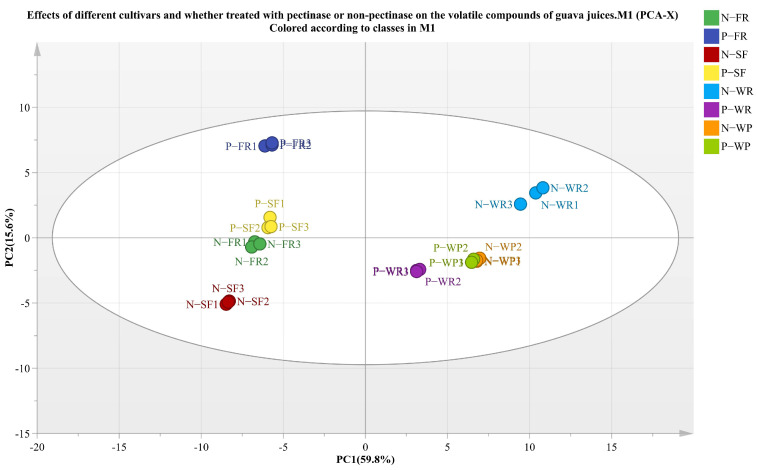
Principal component analysis (PCA) showing the effects of volatile profile of guava juice from different cultivars treated with pectinase or without pectinase.

**Figure 6 foods-12-00330-f006:**
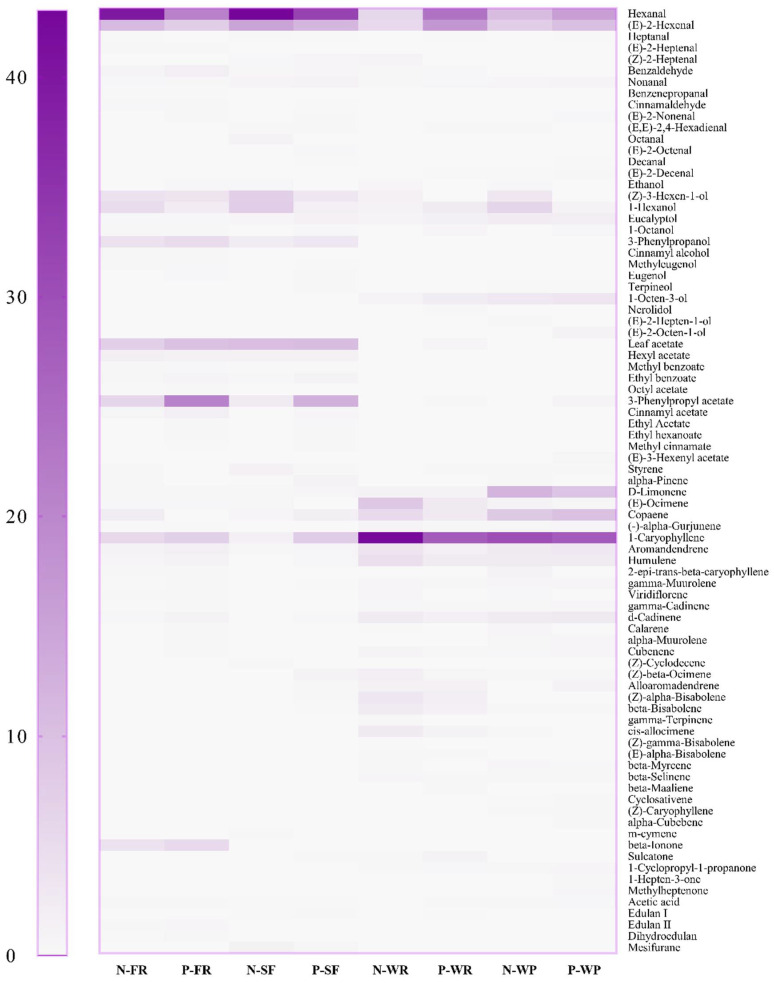
Changes in the relative content of volatile compounds in guava juice from different cultivars treated with pectinase or without pectinase. For the same substance, the more purple its color, the higher its content; the whiter its color, the lower its content.

**Figure 7 foods-12-00330-f007:**
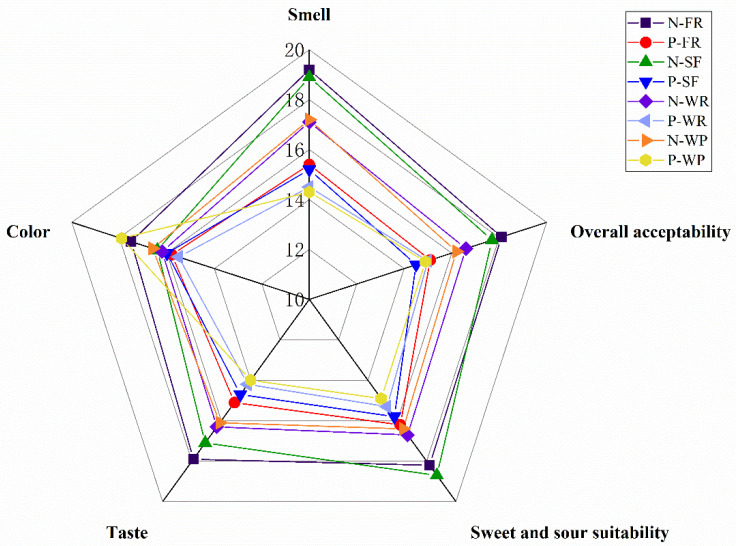
Artificial sensory evaluation radar chart.

**Table 1 foods-12-00330-t001:** Effects of different cultivars and the effect of whether they were treated with pectinase or not on the processing characteristics of guava juice.

Types	Cultivars	Juice Yield (%)	Relative Turbidity (%)	Sedimentation Index (%)	Total Soluble Solids (°Brix)	Titratable Acidity (g/L)
Non-pectinase treatment	FR	65.61 ± 0.36 ^e^	31.96 ± 0.93 ^a^	53.54 ± 0.72 ^a^	9.03 ± 0.02 ^c^	2.70 ± 0.03 ^e^
	SF	60.92 ± 0.21 ^f^	18.67 ± 0.97 ^c^	30.25 ± 0.84 ^b^	7.87 ± 0.02 ^f^	2.53 ± 0.06 ^f^
	WR	67.53 ± 0.31 ^d^	24.33 ± 2.63 ^b^	5.13 ± 0.32 ^d^	7.71 ± 0.01 ^g^	2.65 ± 0.03 ^e^
	WP	64.99 ± 0.22 ^e^	12.92 ± 1.12 ^d^	6.48 ± 0.37 ^c^	8.61 ± 0.02 ^d^	3.54 ± 0.03 ^c^
Pectinase treatment	FR	76.13 ± 0.39 ^a^	13.77 ± 1.12 ^d^	4.97 ± 0.11 ^d^	9.22 ± 0.02 ^b^	3.40 ± 0.03 ^d^
	SF	76.09 ± 0.43 ^a^	12.94 ± 1.01 ^d^	5.42 ± 0.24 ^d^	8.63 ± 0.01 ^d^	3.71 ± 0.03 ^b^
	WR	74.84 ± 0.28 ^b^	8.62 ± 0.29 ^f^	3.74 ± 0.23 ^e^	7.97 ± 0.01 ^e^	3.41 ± 0.03 ^d^
	WP	73.41 ± 0.39 ^c^	12.15 ± 0.49 ^g^	3.41 ± 0.16 ^e^	9.47 ± 0.03 ^a^	4.23 ± 0.04 ^a^

In each column, different letters mean significant differences (*p* < 0.05).

**Table 2 foods-12-00330-t002:** Effects of different cultivars and the effects of whether they were treated with pectinase or not on the color parameters of guava juice.

Types	Cultivars	Color Parameters			Sugar Components (g/L)		
		*L*	a	b	Fructose	Glucose	Sucrose
Non-pectinase treatment	FR	38.05 ± 1.15 ^e^	1.96 ± 0.09 ^b^	1.09 ± 0.27 ^b^	8.88 ± 0.01 ^d^	7.22 ± 0.03 ^g^	16.96 ± 0.62 ^a^
	SF	39.29 ± 0.34 ^e^	−1.74 ± 0.05 ^d^	−1.24 ± 0.09 ^d^	7.95 ± 0.03 ^e^	8.56 ± 0.33 ^e^	14.31 ± 0.03 ^b^
	WR	41.57 ± 0.53 ^d^	2.17 ± 0.40 ^b^	3.23 ± 0.49 ^a^	10.11 ± 0.01 ^c^	9.69 ± 0.08 ^c^	12.31 ± 0.01 ^d^
	WP	47.14 ± 1.15 ^b^	−3.59 ± 0.22 ^f^	3.23 ± 0.35 ^a^	10.16 ± 0.01 ^bc^	9.27 ± 0.04 ^d^	11.79 ± 0.21 ^e^
Pectinase treatment	FR	48.95 ± 0.62 ^a^	−0.57 ± 0.04 ^c^	−1.06 ± 0.06 ^d^	10.20 ± 0.32 ^bc^	7.96 ± 0.26 ^f^	13.73 ± 0.01 ^c^
	SF	47.16 ± 0.19 ^b^	−0.79 ± 0.16 ^c^	−1.81 ± 0.21 ^e^	10.21 ± 0.03 ^bc^	9.42 ± 0.07 ^cd^	12.36 ± 0.10 ^d^
	WR	43.27 ± 0.35 ^c^	3.89 ± 0.10 ^a^	3.06 ± 0.04 ^a^	10.34 ± 0.06 ^b^	10.69 ± 0.09 ^b^	10.20 ± 0.11 ^g^
	WP	46.94 ± 0.90 ^b^	−2.81 ± 0.07 ^e^	0.24 ± 0.28 ^c^	12.13 ± 0.04 ^a^	11.65 ± 0.07 ^a^	11.17 ± 0.04 ^f^

In each column, different letters mean significant differences (*p* < 0.05).

## Data Availability

The data presented in this study are available on request from the corresponding author.

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
