# Peer review of "Effects of Pectinase Pre-Treatment on the Physicochemical Properties, Bioactive Compounds, and Volatile Components of Juices from Different Cultivars of Guava"

_foods, 2023, doi:10.3390/foods12020330_

Round 1

Reviewer 1 Report

Comments and Suggestions for Authors

The authors aimed to study the effects of the cultivars and pectinase treatment on the physicochemical parameters antioxidant capacity and flavor components of the guava juice.

Pectinase catalizes the decomposition of pectin, including pectin esterase and depolymerases. This will result into higher juice yield, but the authors research the flavour change and physicochemical characeristics after this addition. They studied the effects in four cultivars: Small fragrant Fruit (SF), Four-seasons Red (FR), White Pearl (WP), Watermelon Red (WR).

Theu authors found that after pectinase treatment, the main aroma components were significantly reduced.
Therefore they concluded that the primary aroma of the guava juice is dependent on cultivar and the treatment with pectinase.

The manuscript makes a throrough investigation through the parameters influencing yield, physical (sedimentation, turbidity, colour...), and chemical characteristics (acidity, sugar, ascorbic acid, phenol, flavonoids) of the juice, as well as a full sensirial study.
The data treatment and presentation is very well made. Also the statistics used. The manuscript is well written and the findings are supported by the data. Very well explained.

Reviewer 2 Report

Comments and Suggestions for Authors

Recommendation: Major

The manuscript Evaluation and regulation of the physicochemical properties, bioactive compounds, and volatile components of guava juices made from different cultivars, the methodology was reasonable and technically sound.

Comments to the Author:

The main procedure and findings of the study are well expressed. Introduction: A brief survey of existing literature, the purpose, importance, and innovation of the research is well mentioned.

Below are some important suggestions.

Point 1. In the abstract, give the units for total phenolics and flavonoids with their equivalents. For example, 329.52 GAE mg/L

Point 2 It has been stated that the sensory analysis was evaluated over a maximum of 100 points. Which sensory analysis method does this method fall under? If necessary, use citations. At the same time, it seems that the sensory analysis result graph was evaluated over 20 points. I think the method should be called 100 points in total. Revise the sensory analysis method.

Point 3.  Why was the Pearson correlation done only for bioactive compounds? because it would be better if it was also done with color values, ascorbic acid and sugar components. If possible, it would be good to repeat the Pearson correlation analysis.

Point 4. I think the following statement should be removed in the statistical analysis part. I recommend using it in the discussion section.

The Pearson's correlation coefficient measures a linear correlation. In general, the correlation coefficient is 0.8-1.0, it indicates an extremely strong correlation; 0.6-0.8, indicating a strong correlation; 0.4-0.6, indicating moderate correlation; 0.2-0.4, a weak correlation; 0.0-0.2, very weak or no correlation.

Point 5. Increase the dpi of the figures. Also, write the equivalences of the chapters in the whole article. Specify in graphs and tables.

Point 6. The flavor profile results of guava juice should be presented in tabular form. It will be insufficient for us to interpret the results with PCA. Because we cannot see the changes in the aroma profile.

Point 7. No numerical results were given in the sensory evaluation. Therefore, this section should be revised.

Reviewer 3 Report

Comments and Suggestions for Authors

Please see comments in attached file.

Round 2

Reviewer 2 Report

Comments and Suggestions for Authors

It was determined that the authors made the necessary corrections. I think it is suitable for publication.